# Development of a questionnaire to determine the case detection delay of leprosy: A mixed-methods cultural validation study

**Naomi D. de Bruijne**[1,2]*, **Kedir Urgesa**[3,4], **Abraham Aseffa**[3], **Kidist Bobosha**[3],
**Anne Schoenmakers**[2], **Robin van Wijk**[2]*, **Thomas Hambridge**[5], **Mitzi M. Waltz**[1],
**Christa Kasang**[6], **Liesbeth Mieras**[2]

**1** Athena Institute, Faculty of Earth and Life Sciences, VU University, Amsterdam, Netherlands, **2** NLR, Amsterdam, Netherlands, **3** Armauer Hansen Research Institute (AHRI), Addis Ababa, Ethiopia, **4** Haramaya University, College of Health and Medical Sciences, Harar, Ethiopia, **5** Erasmus University Medical Center, Rotterdam, Netherlands, **6** German Leprosy and Tuberculosis Relief Association (DAHW), Wurzburg, Germany

* n.debruijne@nlrinternational.org (NDB); r.vanwijk@nlrinternational.org (RW)

**Data Availability Statement:** The materials developed as part of this study are available in English, Portuguese and Swahili via https://www.

## Abstract

### Background

Delay in case detection is a risk factor for developing leprosy-related impairments, leading to disability and stigma. The objective of this study was to develop a questionnaire to determine the leprosy case detection delay, defined as the period between the first signs of the disease and the moment of diagnosis, calculated in total number of months. The instrument was developed as part of the PEP4LEP project, a large-scale intervention study which determines the most effective way to implement integrated skin screening and leprosy post-exposure prophylaxis with a single-dose of rifampicin (SDR-PEP) administration in Ethiopia, Mozambique and Tanzania.

### Methodology/Principal findings

A literature review was conducted and leprosy experts were consulted. The first draft of the questionnaire was developed in Ethiopia by exploring conceptual understanding, item relevance and operational suitability. Then, the first draft of the tool was piloted in Ethiopia, Mozambique and Tanzania. The outcome is a questionnaire comprising nine questions to determine the case detection delay and two annexes for ease of administration: a local calendar to translate the patient's indication of time to number of months and a set of pictures of the signs of leprosy. In addition, a body map was included to locate the signs. A 'Question-by-Question Guide' was added to the package, to provide support in the administration of the questionnaire. The materials will be made available in English, Oromiffa (Afaan Oromo), Portuguese and Swahili via https://www.infolep.org.

### Conclusions/Significance

It was concluded that the developed case detection delay questionnaire can be administered quickly and easily by health workers, while not inconveniencing the patient. The

leprosy-information.org/resource/case-detection-delay-questionnaire. Data is stored for 25 years according to EU regulation 536/2014 considering clinical medication-related research projects. Data is available in a repository for potential authorized re-use for future data analysis or study replication via https://www.leprosy-information.org/sites/default/files/2021-12/Debruijne%20et%20al.%2C%20CDD%20development_Mergedrawdata.pdf.

**Funding:** This project is part of the EDCTP2 programme supported by the European Union awarded to NLR/LM (grant number RIA2017NIM-1839-PEP4LEP), and the Leprosy Research Initiative (LRI; www.leprosyresearch.org) awarded to NLR/LM (grant number 707.19.58.). The funders had no role in study design, data collection and analysis, decision to publish, or preparation of the manuscript.

**Competing interests:** The authors have declared that no competing interests exist.

instrument has promising potential for use in future leprosy research. It is recommended that the tool is further validated, also in other regions or countries, to ensure cultural validity and to examine psychometric properties like test-retest reliability and interrater reliability.

## Author summary

Leprosy is an infectious disease known for its long incubation period, which contributes to delays in diagnosis that can result in irreversible deformities, mainly of the hands, feet and eyes. To prevent impairments, early case detection followed by prompt treatment of new patients is important. A large-scale intervention study called PEP4LEP was designed to determine the most effective method to implement integrated skin screening and leprosy post-exposure prophylaxis administration with a single-dose rifampicin (SDR-PEP). The main objective of PEP4LEP is to evaluate the effectiveness of two SDR-PEP implementation approaches, by assessing the rate of leprosy patients detected and the case detection delay (CDD) of leprosy in Ethiopia, Mozambique and Tanzania. The CDD is defined as the period between the onset of the first signs of the disease and the diagnosis of leprosy. A standardized tool was needed to measure the CDD in PEP4LEP. Therefore, the aim of this study was to design this methodological tool by conducting a cultural validation study, in which health professionals, leprosy experts, and persons affected by leprosy were consulted. The outcome of this study is a tool, comprising a short questionnaire with two annexes and a 'Question-by-Question Guide', to support the administration of the questionnaire. The materials will be made available in English, Oromiffa (Afaan Oromo), Portuguese and Swahili via https://www.infolep.org.

## Introduction

Leprosy, also called Hansen's disease, is an infectious neglected tropical disease (NTD) known since ancient times, which mainly affects the skin, peripheral nerves and eyes. One of the challenges of the elimination of leprosy lies in its long incubation period during which no signs or symptoms are present but transmission to others is assumed to already take place. The average incubation period is five years, but the presentation of the first obvious signs or symptoms of the disease can take up to 20 years [1].

Since 1982, it has been possible to treat leprosy effectively with multidrug therapy (MDT) [2, 3]. Additionally, since 1995 the World Health Organization (WHO) has provided MDT free of charge for all newly diagnosed patients [3]. Unfortunately, on top of the long incubation period, the diagnosis of leprosy is often delayed because of physical and social barriers, and lack of awareness. These issues contribute to ongoing transmission of the infection to other individuals and also pose risks for the development of irreversible physical impairments [2, 4–8]. For this reason, Smith and colleagues (2014) argue that passive case finding and treatment of patients does not reduce the delay in diagnosis of leprosy effectively, and therefore, will not prevent disability in newly diagnosed patients [9]. Hence, early detection of leprosy cases is crucial to start treatment before permanent disabilities have developed [10, 11].

Since 2018, WHO recommends the administration of single dose rifampicin (SDR) as post-exposure prophylaxis (PEP) for leprosy for contacts of leprosy patients [12]. To determine the most effective method to implement active case finding activities (integrated skin screening) combined with SDR-PEP administration in Ethiopia, Mozambique and Tanzania, a large-scale

intervention study called 'PEP4LEP' was designed [13]. Main outcomes of this project are the rate of leprosy patients detected and the delay in diagnosis. To enable comparison of intervention outcomes in the three East African project countries, it was necessary to develop a standardized methodological tool to determine the case detection delay (CDD) in number of months. The concept of CDD as used in the current study is defined as the period between the onset of the first signs or symptoms of the disease and the diagnosis of leprosy [2, 12, 14]. This period both includes the 'patient delay', which is defined as the duration between notice of the first sign or symptom by the patient to the first health care provider consultation and 'health-system delay', which is the duration between the first health care provider consultation and the moment of leprosy diagnosis [15]. A 'sign' can be described as an objective and observable phenomenon (e.g. by a medical professional), such as a skin patch or a palpable enlarged nerve, whereas a 'symptom' is a subjective experience that can only be identified by the patient themselves, such as pain or a tingling sensation [16, 17]. For reasons of readability of the current article, when mentioned 'signs' in the further text this should be read as 'signs and symptoms'.

Main outcomes of the PEP4LEP project are the rate of leprosy patients detected and the delay in diagnosis [13]. To enable comparison of intervention outcomes in the three East African project countries, it was necessary to develop a standardized methodological tool to determine the CDD in number of months. The concept of CDD as used in the current study is defined as the period between the onset of the first signs of the disease and the diagnosis of leprosy [2, 14].

Ethiopia has the second highest number of annually detected leprosy patients in Sub-Saharan Africa (SSA): 3,201 new cases in 2019 [18]. The number of newly detected patients in Mozambique and Tanzania were 2,220 and 1,603 respectively in 2019. Although the prevalence of leprosy has significantly decreased in these countries since the implementation of MDT, the number of new cases per year has only slightly dropped in the past decade [10]. Ongoing transmission is further suggested by the number of children (<15 years of age) that were diagnosed in 2019: 507 in Ethiopia, 211 in Mozambique, and 53 in Tanzania, of whom 94 presented with grade 2 disabilities in all three countries combined [18]. Consequently, the WHO has identified all three countries as part of the 23 'global priority countries' for leprosy [18].

To assess the CDD of leprosy in these three Sub-Saharan African countries (*Fig 1*), it is necessary to design a methodological tool that is validated and adapted to the cultural context of these countries [19]. Several studies have investigated the CDD of leprosy in countries like Bangladesh [20], Brazil [21–23], India [15], Nepal [24] and Paraguay [25]. However, literature on the delay in diagnosis in the cultural context of Ethiopia, Mozambique and Tanzania is lacking or outdated [26–29]. Although several questionnaires have been designed to determine the CDD of leprosy [23, 24, 30], a specific standardized questionnaire for multi-country use does not exist. Therefore, the objective of this study was to design a methodological tool to determine the delay in diagnosis of leprosy in number of months, that is validated for usage in Ethiopia, Mozambique and Tanzania, and adaptable to other cultural settings.

## Methods

### Ethics statement

Ethical clearance was gained from the Armauer Hansen Research Institute (AHRI) in Addis Ababa, Ethiopia, the ethics board of Lúrio University, Nampula, Mozambique, and from the Catholic University of Health and Allied Sciences (CUHAS) in Mwanza, Tanzania. Informed consent was obtained (written or thumb-printed) from all participants. When including children below 18 years of age, written consent was obtained from a parent / legal guardian. The study guarantees the confidentiality of the content of the data provided by the participants, e.g.

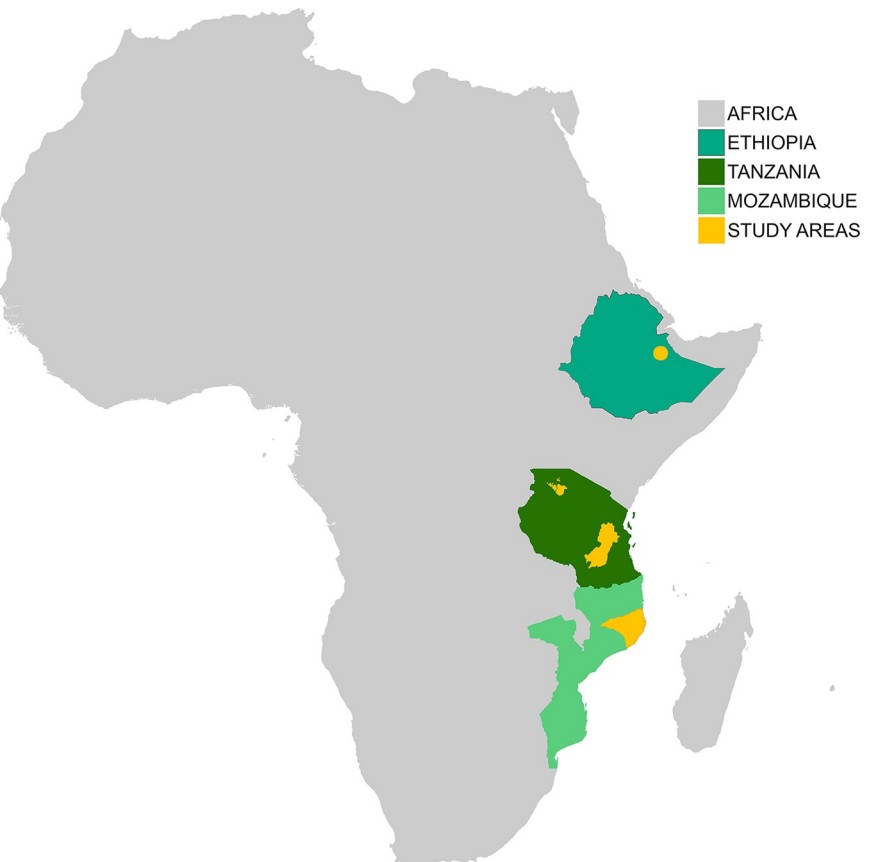

**Fig 1. PEP4LEP study locations–created with ArcGIS Pro. Base layer of the map available via: https://www.arcgis.com/home/item.html?id=64aff05d66ff443caf9711fd988e21dd**

by using numeric codes for pseudonymization. A token of appreciation (e.g. a snack) was given after an interview was conducted.

## Study design

This sub-study was a cross-sectional instrument development and validation study. The study took place in Ethiopia, Bisidimo Hospital in Harar, East Hararghe Zone; in Mozambique in Murrupula Hospital, Nampula Province; and in Tanzania in Morogoro district and in Mwanza district. The study was divided into two phases, the study design is visualised in *Fig 2*.

**Phase 1: Instrument development.** An instrument was developed in phase 1 of the study, which took place in Ethiopia only. A draft questionnaire was developed based on a literature review, complemented with semi-structured interviews with health professionals, a pilot and a focus group discussion (FGD) with persons affected by leprosy and an expert panel with leprosy experts.

**Phase 2: Validation process.** Phase two included three pilot studies in Ethiopia, Mozambique and Tanzania. In Ethiopia, this included test-retest administration of the developed instrument. For this sample, a criterion sampling strategy was used. In Mozambique and Tanzania, a local health worker administered the questionnaire to persons affected by leprosy. Afterwards, these affected persons were interviewed to establish the general validity as well as the cultural validity of the questionnaire. Also, local key informants and health workers were included in additional semi-structured interviews in Mozambique and Tanzania.

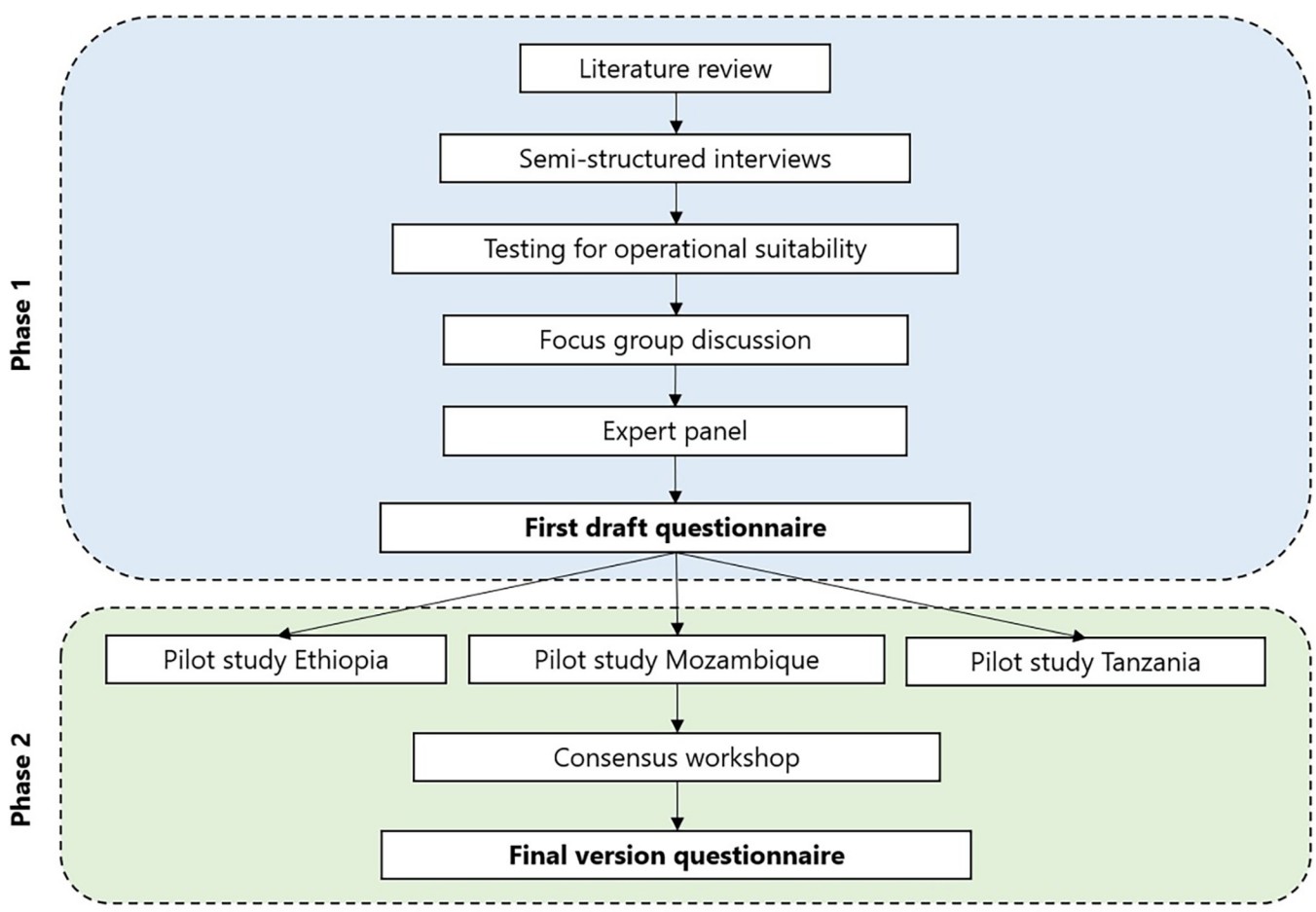

**Fig 2. Visualisation of study design.**

A consensus workshop was organized that included the main researchers of phase 1 and of the three pilot studies. The workshop focused on reaching consensus on a final version of the questionnaire, applicable in the several study settings.

## Study population

The study population comprised of leprosy patients, persons affected by leprosy, local key informants and health workers. In this study, a leprosy patient is defined as someone who is under treatment for leprosy. A person affected by leprosy is cured from the disease, but is living with its consequences such as disability and/or stigma. Details about the inclusion- and exclusion criteria, as well as the sampling strategies are described in *Table 1*. Subjects were eligible to participate if they: 1) were willing to participate, and 2) gave informed consent. There were two exclusion criteria: 1) difficulties in communication, and 2) participants under 18 years old who could not be accompanied by a parent or legal guardian.

## Data analysis

Qualitative data were first transcribed verbatim in the language in which data collection was done. Non-English transcripts were translated into English by a professional translator, a native speaker of the local language. Then, the transcripts were coded using ATLAS.ti 8.3.20.0

**Table 1. In-, exclusion criteria and sampling strategy per sample.**

| Phase | Sample | Study item | Group | Inclusion criteria | Exclusion criteria | Sampling strategy |
|---|---|---|---|---|---|---|
| 1 | 1 | Semi-structured interviews | Health professionals | Employment within East Hararghe Zone; familiar with the cultural context of the region; familiar with leprosy; willing to participate; willing to provide informed consent. | Unable to speak or understand the English language. | Purposive sampling |
| 1 | 2 | Questionnaire: Operational suitability | Leprosy patients | Affected by leprosy; diagnosed and/or treated in Bisidimo Hospital; willing to participate; willing to provide informed consent. | Living with a disability that hampers participation (e.g. deafness); younger than 18 and could not be accompanied by a legal guardian; diagnosed with leprosy longer than six months ago. | Typical case sampling |
| 1 | 3 | Focus group discussion | Persons affected by leprosy | Affected by leprosy; hospitalized in BH; older than 18 years old; willing to participate; willing to provide informed consent. | Living with a disability that hampered participation (e.g. deafness). | Typical case sampling |
| 1 | 4 | Expert panel | Experts in the field of leprosy | Experience in the field of leprosy | Unable to speak or understand the English language. | Purposive sampling |
| 2 | 5 | Ethiopia: Questionnaire's pilot | Leprosy patients | Leprosy diagnosis confirmed by a dermatologist after physical examination and laboratory tests; diagnosed and/or treated in Bisidimo Hospital; willing to provide informed consent. | Living with any health condition interfering with their memory or other mental abilities; younger than 18 and could not be accompanied by a legal guardian; diagnosed with leprosy longer than six months ago. | Convenience sampling |
| 2 | 6 | Mozambique: Questionnaire's pilot | Leprosy patients | Willing to participate; willing to provide informed consent. | Younger than 18 and could not be accompanied by a legal guardian; not yet diagnosed with leprosy; diagnosed with leprosy longer than six months ago. | Convenience sampling |
| 2 | 7 | Mozambique: Semi-structured interviews | Healthcare workers | Working in Murrupula Hospital; familiar with leprosy; willing to participate; willing to provide informed consent. | *None* | Purposive sampling |
| 2 | 8 | Tanzania: Questionnaire's pilot | Leprosy patients | Residing in Morogoro region; familiar with the cultural context; diagnosed with leprosy; willing to provide informed consent. | Living with a disability that hampers participation (e.g. deafness); younger than 18 and could not be accompanied by a legal guardian; diagnosed with leprosy longer than six months ago. | Purposive sampling |
| 2 | 9 | Tanzania: Semi-structured interviews | Key informants/ healthcare workers | Fluent in Swahili; familiar with the cultural context of Morogoro region; familiar with leprosy; able to understand the purpose of the study; willing to provide informed consent. | Living with a disability that hampers participation (e.g. deafness); younger than 18 and could not be accompanied by a legal guardian; diagnosed with leprosy longer than six months ago. | Purposive sampling |

(Berlin: ATLAS.ti Scientific Software Development GmbH). Themes were identified following the components of the conceptual framework of Herdman et al. [19]. Related codes and themes were identified by axial coding, and sub-themes were identified [31]. Quantitative data was analyzed using descriptive statistics derived from Microsoft Office Excel.

## Conceptual framework

The research design was based on the conceptual framework of Herdman et al. [19]. As used in the current study, the framework consists of five components: conceptual understanding, item relevance, semantic understanding, operational suitability and measurement consistency, which are summarized in 'functional equivalence.' Based on the argumentation in the systematic review of Stevelink and van Brakel (2013), the component of functional equivalence is translated in this conceptual framework as 'cultural validity' [19, 32].

## Results

The review of the available literature resulted in six relevant questionnaires that were previously used to determine the CDD of leprosy Brazil [23], India [15], Nepal [24], and Sierra Leone [30], and to determine the CDD of tuberculosis in East Hararghe Zone [33] and Ethiopia [34]. Aids applied in questionnaires in the cultural context of Ethiopia were identified in three studies, including the use of a local calendar [35], and seasonal and religious festivals or events [36, 37]. Furthermore in some studies pictures or cards were used as visual support for the tool/questionnaire [38, 39]. In the initial interviews (Sample 1), relevant parts of the questionnaires, as well as potential annexes, were discussed. Based on this, the following items were included 1) introduction and instructions, 2) questions regarding CDD, 3) annexes to support administration of the questionnaire, and 4) a sheet to calculate the CDD.

An overview of general characteristics of the study population per study sample can be found in *Table 2*. Sample 1 included six participants in professions such as (senior) general practitioner, nurse and dermato(venero)logist. Years of working experience varied from 6 to 15 years. Sample 7 consisted of 5 healthcare workers with 5 to 10 years of work experience. Sample 9 consisted of three key informants including a leprosy specialized physician, the head of the national leprosy and TB programme, a general practitioner (GP) experienced in dermato(venero)logy, and six healthcare workers who had experience with leprosy patients. Furthermore, the expert panel (Sample 4) consisted of researchers in the field of leprosy. In total, 89 persons affected by leprosy were included in the study. Of these, 15 participated in study phase 1 and 74 participated in phase 2. Among these samples, 68 to 100% of the participants were residents from rural areas. Older leprosy patients in these areas often lacked any formal education and were illiterate, which must be taken into account in the operational suitability of the methodological tool.

**Table 2. Overview of general characteristics of the study population per study sample.**

| Phase | Sample | Study item | Population | N | Sex | Age range (years) |
|---|---|---|---|---|---|---|
| 1 | 1 | Semi-structured interviews | Health professionals | 6 | M: 6 F: 0 | 29–62 |
| 1 | 2 | Questionnaire: Operational suitability | Leprosy patients | 8 | M: 8 F: 0 | 18–70 >* |
| 1 | 3 | Focus group discussion | Persons affected by leprosy | 7 | M: 5 F: 2 | < 25–70 >* |
| 1 | 4 | Expert panel | Experts in the field of leprosy | 3 | M: 3 F: 1 | 60+ |
| 2 | 5 | Ethiopia: Questionnaire's pilot | Leprosy patients** | 49 | M: 35 F: 14 | 18–50 >* |
| 2 | 6 | Mozambique: Questionnaire's pilot | Leprosy patients** | 18 | M: 10 F: 8 | 10–76 |
| 2 | 7 | Mozambique: Semi-structured interviews | Healthcare workers | 5 | M: 4 F: 1 | 28–42 |
| 2 | 8 | Tanzania: Questionnaire's pilot | Leprosy patients** | 7 | M: 3 F: 4 | 23–75 |
| 2 | 9 | Tanzania: Semi-structured interviews | Key informants / healthcare workers | 9 | M: 6 F: 3 | 27–59 |

*Age was recorded in age groups

** In some cases (~6%), diagnosis had occurred more than 6 months prior to the study

## Conceptual understanding

**Time.**   All health professionals in Sample 1 agreed that patients from rural areas struggle to express time, and thus, the duration of leprosy signs. This may be because they cannot remember the onset of signs, or because they do not use a calendar or use solely religious calendars. Health professionals in Sample 1 said that patients from rural areas sometimes explain the duration of their signs in years and months. However, they more frequently use (agricultural) seasons to express the duration of signs. This was confirmed in the pilot projects in Tanzania and Mozambique, where agricultural periods were used as references to remember the onset of signs.

In addition, one participant specifically mentioned that it is important to check a patient's answer, as *"they may tell you different things"* (Interview, Sample 1, leprosy researcher). This issue was also raised by the expert panel in Sample 4, where the importance of double-checking the patient's answers in a questionnaire was stressed by multiple leprosy experts.

**Leprosy.**   A recurrent issue mentioned, in both interviews and the FGD, is that misconceptions about the transmission of leprosy contribute to the delay in diagnosis. As one participant said:

> *"Before visiting a health worker we don't know anything about the disease"* (FGD, Sample 3, woman).

Although patients may struggle to recognize the signs of leprosy, health professionals in Sample 1 explained that patients do not struggle to describe their disease progression. This also emerged from the FGD, where each participant described their disease progression in a detailed manner.

A 'wait and see' mentality amongst patients was brought up regularly by health professionals in Sample 1: patients do not undertake any steps, expecting that the signs will resolve naturally. Other patient journeys that were mentioned by Sample 1 and Sample 3 included self-treatment, which involved traditional or religious treatments, or heating the skin with fire. All health professionals in Sample 1 agreed that stigma in the context of leprosy is caused by a lack of education and incorrect knowledge about the disease. In the FGD with persons affected by leprosy, a number of expressions addressing stigma and discrimination were brought up, for instance regarding social exclusion. Participants explained that they were not respected by others, forced to leave their communities and/or excluded from wedding ceremonies. These are all explanations of reasons why patients are delaying their visit to the health centre. Health professionals in Sample 1 also mentioned that patients often go to health facilities where they are misdiagnosed, causing a health system delay.

## Item relevance

**Questionnaire questions.**   The item relevance of the signs of leprosy in the questionnaire and the picture set was discussed with health professionals from Sample 1 and consensus was reached on the items to be included in the questionnaire. These items were checked, and reformulated if needed after discussion with local professionals and the expert panel in Sample 4. Options regarding the question on the relevant steps that were taken by before diagnosis, as discussed by health professionals in Sample 1, were confirmed during the FGD and after the pilot projects. After the consensus workshop, it was decided to formulate these two questions as open questions, to prevent cueing of answers. Relevant items were still listed underneath the questions as examples to assist the assessor. The item relevance of reasons for delayed diagnosis were discussed with the health professionals from Sample 1. In the first draft of the

questionnaire, this question was formulated as a closed 'list question.' However, after the consensus workshop, it was decided to focus this question solely on health system delay and to formulate two open questions: *'When was your first visit to a health facility?'* and *'How many times did you visit a health facility before you received your diagnosis?'*.

**Picture set.** During the interviews with health professionals in Sample 1, two aids for administration of the questionnaire were discussed. The first was a set of pictures with signs of leprosy. All health professionals agreed that this would be helpful in administering the questionnaire: they can be used to help patients to recognize signs, increase patient knowledge, and potentially help patients' recall the onset of signs. For the picture set, pictures of leprosy signs on dark skin were selected to match the skin colour of the general SSA population.

When the first draft of the questionnaire was tested for operational suitability with Sample 2, it was observed that the picture set was frequently used during administration of the questionnaire. Also, during the pilot study in Tanzania, the picture set was seen as "*really helpful*":

> *"Most of our patients fail to explain [their disease]. So, through these pictures they can tell us: 'Ah it was like this', or: 'Oh it was like that.'"* (Pilot study, Sample 9, key informant).

Pictures were derived from several sources, of which 'A New Leprosy Atlas', the 'International Federation of Anti-Leprosy Associations (ILEP) learning guide on how to diagnose and treat leprosy', the NLR SkinApp, and Google Images were mainly used [40, 41].

**Local calendar.** The health professionals in Ethiopia, Mozambique and Tanzania explained that the use of a calendar might help them administering the questionnaire, and help patients to remember the onset of their leprosy signs. The health professionals in Sample 1, 7 and 9 agreed that the calendar should be designed in the local language, and that it should contain seasons and local agricultural events. Additionally, the expert panel indicated that big (political, environmental, sport-related) events of the past five years could be helpful to indicate the CDD and should therefore be included in the calendar. This method was also described in literature [36]. In Ethiopia, the local calendar was created by combining the solar calendar that is nationally used (the Ethiopian Calendar is 7–8 years behind the Gregorian calendar, comprising of 13 months per year of which the last month counts 5–6 days) with agricultural seasons [42, 43]. In Tanzania, the content of the local calendar was extended by adding the agricultural seasons for different types of crops. Also, the list of major events was extended by including presidential elections and extreme weather events. In the pilot study with leprosy patients in Mozambique (Sample 6), the local 'farming' calendar was perceived as helpful by participants. Furthermore, in Tanzania it was found useful to ask the patient when personal experiences (e.g. birth of a child, new job) took place in order to recall timing of the first leprosy signs. When the questionnaire was tested with Sample 2 (Ethiopia), the local calendar was successfully used to understand the patient's indication of time and to translate it to number of months.

**Body map.** To indicate the location of the first sign of leprosy on the patient's body, and to minimize recall problems, a body map was included in the questionnaire. On this body map, the health worker administering the questionnaire can mark the location where the first sign of leprosy was noticed. The body map was included after the consensus workshop and was not tested in the pilot studies of Phase 2.

## Semantic understanding

Forward translation from English to the local language was checked during the interviews with persons affected by leprosy. Words that were perceived stigmatizing were omitted and more

neutral words were included. After back translation of the questionnaire, items that showed overlap in the back translation (for instance, "*because I am afraid/embarrassed*" and "*I am embarrassed because it was leprosy*") were merged. Other questions were removed or specified.

## Operational suitability

All health professionals from Sample 1 confirmed that the questionnaire should be administered in a face-to-face interview-like manner, as patients are often illiterate. Five health professionals of Sample 1 said that the best time for administration of the questionnaire would be right after diagnosis, as patients may be lost to follow-up or referred to local hospitals to receive MDT. Two participants specifically took the emotional status of patients directly after the diagnosis is shared into account, but still concluded that it was best to administer the questionnaire at this point in the patient journey. These findings were confirmed when the questionnaire's first draft was tested for operational suitability: the administration of the questionnaire did not inconvenience any participant of Sample 2.

The CDD Questionnaire will be available in the languages used in the PEP4LEP project, including English, Oromiffa (Afaan Oromo), Portuguese and Swahili.

**Question-by-Question Guide.** To support consistent CDD questionnaire administration between researchers, a 'Question-by-Question Guide' was designed at the end of the questionnaire development phase [44]. This guide provides explanations on the aim of the 10 questions asked in the questionnaire, 'prompt questions' which can be used if study participants do not understand the original questions, questionnaire administration tips and examples of possible answers given by study participants.

## Measurement consistency

During the pilot study in Ethiopia of the questionnaire (Phase 2), the majority of respondents reported that the onset of the disease was between one and two years before the diagnosis. The relation of the CDD of leprosy patients to the disability grade at the moment of diagnosis is shown in the boxplots in *Fig 3*. The answers given to the questions '*When did you see the first signs of your disease*?' and '*Can you explain how long you have had these signs*?' were inconsistent, as respondents reported two different moments of onset of the disease. Overall, the answers to the two questions were consistent for 28 patients. In the remaining 18 cases, when inquiring about signs individually, additional or different signs were listed during the moment of onset of the first sign or before. In 11 cases this led to a discrepancy in the number of months ago that the first sign was experienced, with an on average difference of 9.5 months.

To five patients in Sample 5, the CDD questionnaire was administered twice by the same interviewer with three days apart. For one patient the case detection delay outcome differed between the two moments of interviewing by two months. *Table 3* shows the count of the matching and non-matching answers between the two subsequent administrations to the same five participants. The non-matching responses were either because patients reported different times of onset of the disease and/or of specific signs in the two administrations, or because certain signs that were reported as experienced in the first administration were not in the second one. Notably, the answers provided in regard to the time of onset of enlarged nerves did not match for any of the repeated administrations of the questionnaire.

## Discussion

The objective of this study was to design a questionnaire to determine the CDD of leprosy in number of months. While designing the methodological tool, cultural validity was ensured by investigating conceptual understanding, item relevance, semantic understanding, operational

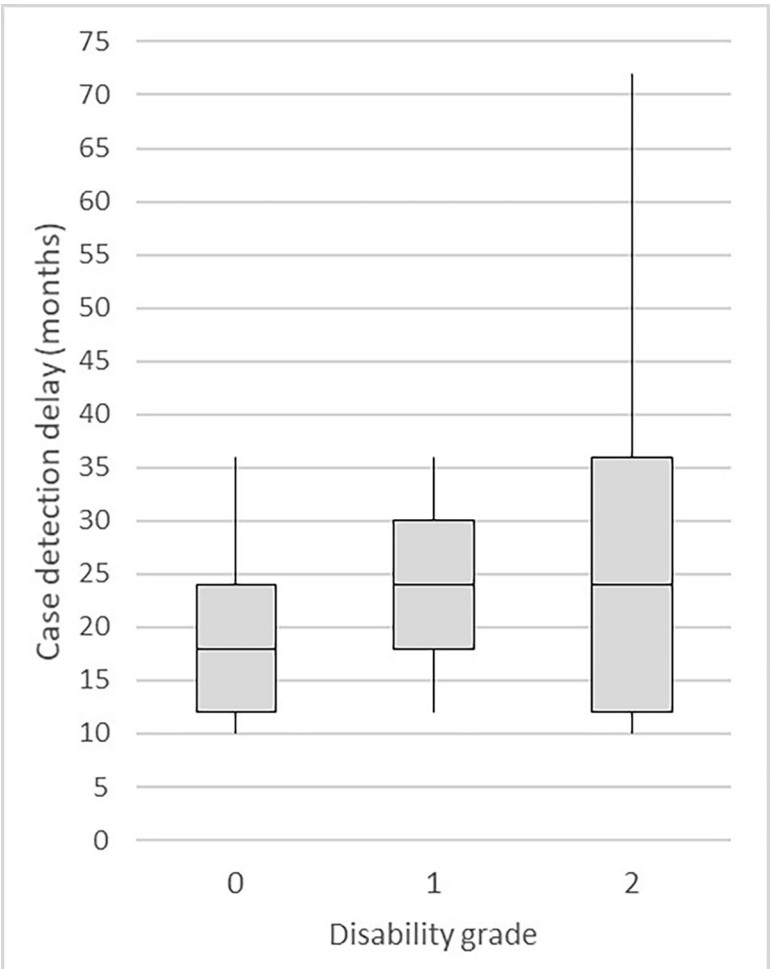

**Fig 3. Case detection delay in relation to the disability grade at the moment of diagnosis, leprosy patients in Ethiopia, Sample 5 (n = 49).**

**Table 3. Count of matching / non-matching responses in test-retest validation.**

| | Matching (n) | Non-matching (n) |
|---|---|---|
| When did you see the first signs of your disease? | 4 | 1 |
| Which sign was it that you saw first? | 2 | 3 |
| Can you explain how long you have had these signs? | | |
| 1. Skin lesions with loss of sensation | 3 | 2 |
| 2. Nodules | 5 | - |
| 3. Enlarged nerves | - | 5 |
| 4. Numbness | 2 | 3 |
| 5. Foot drop | 5 | - |
| 6. Wrist drop | 5 | - |
| 7. Claw hands | 4 | 1 |
| 8. Painless wounds | 5 | - |
| 9. Reabsorbed fingers | 5 | - |
| 10. Reaction | 3 | - |
| **Total** | 47 | 16 |

suitability and measurement consistency of the tool in the cultural context of three East-African countries: Ethiopia, Mozambique and Tanzania. Insight in the previously mentioned themes has led to the design of questionnaires that can be used to determine the CDD of leprosy in the three study countries. The questionnaire includes guidelines on how to administer the questionnaire and two annexes (a local calendar, picture set) to support administration of the questionnaire. The questionnaire's administration duration is approximately 15 minutes.

Unfortunately, previous research does not elaborate on either the item relevance or the semantic understanding of questionnaires on CDD. Little knowledge is available regarding the operational applicability of such questionnaires in similar contexts, which hampers comparability of outcomes of this study. In accordance with outcomes of the present study however, in previous studies similar questionnaires were used in East Hararghe Zone, administered by trained local data collectors [45–48]. Although all health professionals in Sample 1 agreed on the preferred moment of questionnaire administration (i.e., right after diagnosis, to prevent loss to follow-up), this is not in line with previous findings. For instance, Henry et al. (2016) proposed administering the questionnaire at least one month after the diagnosis, to prevent emotional distress. But it should be considered that this study was conducted in tertiary referral centres in Brazil (23). The best moment for administration may depend on the logistics of the leprosy control services in the country.

In literature, there seems to be no consensus on a reliable period for people to recall health related events, and periods range from days or weeks to years [49, 50]. However, recalling the existence of signs seems to be easier than recalling for example pain intensity and quality of life [49]. Major signs or 'high impact conditions' as well as more unusual signs are recalled better than minor signs/'low impact conditions' or more commonly occurring signs [49, 51]. In leprosy research, it could therefore be the case that late signs (e.g., paralysis or ulcer) are easier recalled than early signs (e.g., painless rash / single skin lesion) because of impact and severity. Amjadi et al. (2004) stated that the accuracy of recall of the onset of rheumatoid arthritis symptoms tends to decline over a five-year period [52]. When administering our questionnaire, we advise to include patients who were diagnosed up to six months back, and who are usually still using MDT to treat their leprosy, to minimize potential problems of recall [53]. A calendar was added as a supplement to support the accuracy of recalling the first signs [8, 36]. For each cultural setting, an appropriate calendar should be identified/designed and included, before implementing the questionnaire.

## Strengths & limitations

This study makes a noteworthy contribution to the field of leprosy research by designing a methodological tool to determine the CDD of leprosy in number of months. In line with this, the multi-faceted, methodical study design which included pilot projects in three countries in East Africa is considered a strength of this study.

On the other hand, several limitations need to be considered. An important limitation of this study is the small sample sizes. Considering the low number of recently diagnosed leprosy patients per health centre, a prolonged field study is necessary to assess the measurement consistency (i.e., test-retest reliability and inter-rater reliability) of the questionnaire. Additionally, the current study might have been limited by language barriers, as the health professionals included as well as the researchers were communicating in their second or third language. Though the included picture set and body map can help to at least avoid the usage of medical jargon [50]. Furthermore, Stull et al. (2009) stated in their review on 'optimal recall periods for patient-reported outcomes,' that scientists must take factors that can influence recall into account during instrument and trial development [49]. For example, minimizing the

complexity of questions is vital. In this questionnaire, repetitive questioning was introduced to optimize recall. Adding context can also help to prevent potential recall problems [49, 50].

For future research, further validation of the questionnaire, especially measurement consistency (inter-rater reliability, test-retest) is advised. Also, when using the questionnaire in other countries or cultural contexts, adapting (e.g., local language. pictures with other skin colour, different calendar) and re-validating the questionnaire should be considered.

## Conclusion

This study has led to the development of a questionnaire to determine the CDD of leprosy in the context of Ethiopia, Mozambique and Tanzania. A standardized version was made in English by removing all culturally specific information to enable adaptation to other cultural contexts. The questionnaire, including annexes and Question-by-Question Guide, can be requested by contacting the corresponding author. English, Oromiffa (Afaan Oromo), Portuguese and Swahili language versions of the questionnaire will also be digitally published on https://www.infolep.org [54].

Further research into the measurement consistency of the questionnaire is recommended before implementation in future projects.

## Acknowledgments

The Ethiopian team from AHRI/ALERT and Haramaya University for facilitating and supporting data collection in the field. The study participants, Dr. Alene and Mr. Abrahim. The leprosy experts for their contribution in the expert panel: Paul Saunderson, Ulla-Britt Engelbrektsson and Wim van Brakel. Valerie van der Meij, Adele Barlassina and Marije Tawil for conducting the pilot studies in Tanzania, Ethiopia and Mozambique. The PEP4LEP project team.

## Author Contributions

**Conceptualization:** Naomi D. de Bruijne, Kedir Urgesa, Abraham Aseffa, Kidist Bobosha, Anne Schoenmakers, Thomas Hambridge, Mitzi M. Waltz, Christa Kasang, Liesbeth Mieras.

**Data curation:** Naomi D. de Bruijne, Kedir Urgesa.

**Formal analysis:** Naomi D. de Bruijne, Anne Schoenmakers.

**Funding acquisition:** Christa Kasang, Liesbeth Mieras.

**Investigation:** Naomi D. de Bruijne, Kedir Urgesa.

**Methodology:** Naomi D. de Bruijne, Abraham Aseffa, Kidist Bobosha, Anne Schoenmakers, Robin van Wijk, Mitzi M. Waltz, Liesbeth Mieras.

**Project administration:** Kedir Urgesa, Abraham Aseffa, Kidist Bobosha, Anne Schoenmakers, Robin van Wijk, Christa Kasang, Liesbeth Mieras.

**Resources:** Christa Kasang, Liesbeth Mieras.

**Supervision:** Kedir Urgesa, Abraham Aseffa, Kidist Bobosha, Anne Schoenmakers, Robin van Wijk, Thomas Hambridge, Mitzi M. Waltz, Liesbeth Mieras.

**Validation:** Kidist Bobosha, Anne Schoenmakers, Robin van Wijk, Thomas Hambridge, Mitzi M. Waltz, Liesbeth Mieras.

**Visualization:** Naomi D. de Bruijne, Anne Schoenmakers, Robin van Wijk.

**Writing – original draft:** Naomi D. de Bruijne, Kedir Urgesa, Anne Schoenmakers, Robin van Wijk.

**Writing – review & editing:** Naomi D. de Bruijne, Kedir Urgesa, Abraham Aseffa, Kidist Bobosha, Anne Schoenmakers, Robin van Wijk, Thomas Hambridge, Mitzi M. Waltz, Christa Kasang, Liesbeth Mieras.

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
