## [Decision Letter · Decision Letter 0]

1 Sep 2021

Dear Ms, van Wijk,

Thank you very much for submitting your manuscript "Development of a questionnaire to determine the case detection delay of leprosy: A cross-sectional mixed-methods cultural validation study" for consideration at PLOS Neglected Tropical Diseases. As with all papers reviewed by the journal, your manuscript was reviewed by members of the editorial board and by several independent reviewers. In light of the reviews (below this email), we would like to invite the resubmission of a significantly-revised version that takes into account the reviewers' comments. 

We cannot make any decision about publication until we have seen the revised manuscript and your response to the reviewers' comments. Your revised manuscript is also likely to be sent to reviewers for further evaluation.

Sincerely,

Alberto Novaes Ramos Jr

Associate Editor

Godfred Menezes

Deputy Editor

Reviewer's Responses to Questions

**Key Review Criteria Required for Acceptance?**

**Methods**

-Are the objectives of the study clearly articulated with a clear testable hypothesis stated?

-Is the study design appropriate to address the stated objectives?

-Is the population clearly described and appropriate for the hypothesis being tested?

-Is the sample size sufficient to ensure adequate power to address the hypothesis being tested?

-Were correct statistical analysis used to support conclusions?

-Are there concerns about ethical or regulatory requirements being met?

Reviewer #1: Methods are clearly articulated. Researchers clearly state important limitations of study: Language, Small sample size, needs for further test-retest reliability and inter-rater reliability on the questionnaire. Strengths are that it was developed and tested in 3 countries in East Africa.

Reviewer #2: Please see my comments in the article

Reviewer #3: 1) Objectives of the study are clearly defined. Suggestion to add appropriate key words.

2) The study design is appropriate. Suggestion to add abbreviations.

3) When describing the population the inclusion and exclusion criteria might differ for each of the groups included in the study. So I would suggest to elaborate or add a box/table for clarity.

4) I would suggestion authors to describe the sample size for each category of the persons included in the study: people affected with leprosy, health care workers, key informants and the researchers.

**Results**

-Does the analysis presented match the analysis plan?

-Are the results clearly and completely presented?

-Are the figures (Tables, Images) of sufficient quality for clarity?

Reviewer #1: Table 2 needs correcting(See below in Editorial section)

Include final standardized version of the questionnaire for the reader with a link to the guidelines for administrating it.

Reviewer #2: Please see my comments in the article

Reviewer #3: 1) The analysis presented does match the analysis plan.

2) On the results, it would be interesting to know the grade of disability to correlate with the CDD and could be an indicator of the sensitivity of the questionnaire.

3) Format: Table 2: Lines one and two are the same. Please check.

**Conclusions**

-Are the conclusions supported by the data presented?

-Are the limitations of analysis clearly described?

-Do the authors discuss how these data can be helpful to advance our understanding of the topic under study?

-Is public health relevance addressed?

Reviewer #1: Final Standardized Questionnaire and guidelines were not available to reviewer. It was stated that a standardized version was made in English removing culturally specific information enabling it to be adapted to other cultural settings but it was not possible for the reviewer to observe this.

Reviewer #2: Please see my comments in the article

Reviewer #3: 1) It will be useful to read the standard English questionnaire (not included in the PDF file).

**Editorial and Data Presentation Modifications?**

Reviewer #1: Minor Revisions: 

Table 2: When did you see the first signs of your disease was repeated twice

Alignment within References needs to be corrected. It changes after 12 on line 522.

Include Final CDD Questionnaire and link to Guidelines.

Reviewer #2: Please see my comments in the article

Reviewer #3: Minor Revisions

**Summary and General Comments**

Reviewer #1: This hopefully will standardize the methods to measure case detection delay which will allow comparison across cultures and countries. It is important to do test -retest reliability and inter-rater reliability testing in countries where developed as well in other countries.

Reviewer #2: Please see my comments in the article

Reviewer #3: This is an important study as delay in diagnosis of leprosy is the key for the control and prevention of transmission as stated in the Global Leprosy Strategy 2021-30. As there is no clear signs for the disease onset, like fever or cough for TB. Assessing the case detection delay with a validated questionnaire is the key to aim for the early diagnosis followed by initiation of treatment.

PLOS authors have the option to publish the peer review history of their article (what does this mean?). If published, this will include your full peer review and any attached files.

Reviewer #1: Yes: Linda Faye Lehman

Reviewer #2: No

Reviewer #3: Yes: Ashish Nareshkumar Wagh
---

## [Decision Letter · Decision Letter 1]

27 Nov 2021

Dear Ms, van Wijk,

We are pleased to inform you that your manuscript 'Development of a questionnaire to determine the case detection delay of leprosy: A mixed-methods cultural validation study' has been provisionally accepted for publication in PLOS Neglected Tropical Diseases.

Best regards,

Alberto Novaes Ramos Jr

Associate Editor

Godfred Menezes

Deputy Editor

Reviewer's Responses to Questions

**Key Review Criteria Required for Acceptance?**

**Methods**

-Are the objectives of the study clearly articulated with a clear testable hypothesis stated?

-Is the study design appropriate to address the stated objectives?

-Is the population clearly described and appropriate for the hypothesis being tested?

-Is the sample size sufficient to ensure adequate power to address the hypothesis being tested?

-Were correct statistical analysis used to support conclusions?

-Are there concerns about ethical or regulatory requirements being met?

Reviewer #2: Acceptable

Reviewer #3: Methods are clearly articulated.

Suggested abbreviations were added.

Study population described in table 1.

Sample size well described in table 2.

**Results**

-Does the analysis presented match the analysis plan?

-Are the results clearly and completely presented?

-Are the figures (Tables, Images) of sufficient quality for clarity?

Reviewer #2: OK

Reviewer #3: Case detection delay in relation to the disability grading at the time of diagnosis explained in table 3.

**Conclusions**

-Are the conclusions supported by the data presented?

-Are the limitations of analysis clearly described?

-Do the authors discuss how these data can be helpful to advance our understanding of the topic under study?

-Is public health relevance addressed?

Reviewer #2: OK

Reviewer #3: Conclusions are supported by data.

Public health relevance noted.

**Editorial and Data Presentation Modifications?**

Reviewer #2: Accept

Reviewer #3: (No Response)

**Summary and General Comments**

Reviewer #2: Overall the study conclusions are in lines with the stated objectives

Reviewer #3: This study help the research community to standardize the methods to measure case detection delay. This study can be further extended and can be adopted in other cultural settings and countries.

PLOS authors have the option to publish the peer review history of their article (what does this mean?). If published, this will include your full peer review and any attached files.

Reviewer #2: **Yes: **SRINIVAS GOVINDARAJULU

Reviewer #3: No

---

## [Editor Report · Acceptance letter]

30 Dec 2021

Dear Ms, van Wijk,

We are delighted to inform you that your manuscript, "Development of a questionnaire to determine the case detection delay of leprosy: A mixed-methods cultural validation study," has been formally accepted for publication in PLOS Neglected Tropical Diseases.

Best regards,

Shaden Kamhawi

co-Editor-in-Chief

Paul Brindley

co-Editor-in-Chief
